# BREAKING THE BIAS: QUANTIFYING THE ATTENTION OF INDUSTRIAL ANOMALY DETECTION

## ABSTRACT

The method based on reconstruction and discrimination has made significant progress in unsupervised industrial anomaly detection (IAD) by using generative models to accurately reconstruct normal regions while exhibiting reconstruction failures in anomalous areas. However, current methodologies present two primary limitations. First, reliance on synthetic anomalies and reconstruction loss metrics introduces inadequate supervisory guidance for targeted model optimization. Second, uniform optimization strategies applied indiscriminately across all image regions neglect spatial discrepancies in model confidence levels. We propose **P**seudo-**L**abel **S**upervision in Unsupervised Industrial **A**nomaly **D**etection (PLSAD), a novel framework integrating unsupervised learning with pseudo-label supervision. Our methodology focuses on the differences between the original images and the synthetic anomaly images,thereby decoupling reconstruction processes from discriminative feature learning. This dual-stream architecture not only enhances feature representation robustness but also mitigates error propagation through explicit separation of learning objectives. Furthermore, we introduce **A**daptive **I**ntersection-over-Union **W**eighting (AIW), which dynamically evaluates the model's local performance through pseudo-label and synthetic ground truth alignment, and automatically emphasizes challenging regions. Comprehensive experiments on three IAD benchmarks (MVTec-AD, MVTec-LOCO, VisA) confirm PLSAD's competitive performance in both detection accuracy and anomaly localization.

## 1 INTRODUCTION

Anomaly detection aims to identify and localize data instances that deviate from normal observations. Due to the high cost of labeling and the unpredictable nature of defects, unsupervised deep learning-based anomaly detection methods have rapidly advanced. These methods are trained exclusively on normal samples. Reconstruction-based approaches (such as autoencoders Akcay et al. (2018) and generative adversarial networks Schlegl et al. (2019)) exploit their capacity to reconstruct normal patterns and utilize reconstruction errors to detect anomalous regions, rendering them mainstream solutions. However, their reliance on unsupervised learning restricts their capability to detect subtle and hard-to-identify anomalies, as they lack explicit supervisory signals necessary for optimizing discriminative feature learning. This paper proposes a Pseudo-Label Supervised Anomaly Detection (PLSAD) mechanism that introduces pseudo-labels into unsupervised training, thereby bridging the performance gap between unsupervised and fully supervised methods.

The advanced reconstruction-based method employs a dual-network architecture: a reconstruction sub-network rectifies synthetic anomalies, while a discrimination sub-network segments the anomalies by comparing the original and reconstructed images. Despite their considerable effectiveness, these methods exhibit certain limitations: 1. Single training perspective: The discriminative model frequently employs concatenated synthetic anomaly images and reconstructed images as input during training, but the reconstructed images are not the "standard references." This singular loss formulation is often insufficiently comprehensive, resulting in a "lack of drive" in the later stages of training. 2. Neglect of model confidence: Existing methods uniformly treat all regions during training, disregarding the spatial variability in reconstruction quality. Regions with poor reconstruction require stronger supervision to optimize decision boundaries, whereas excessive supervision in

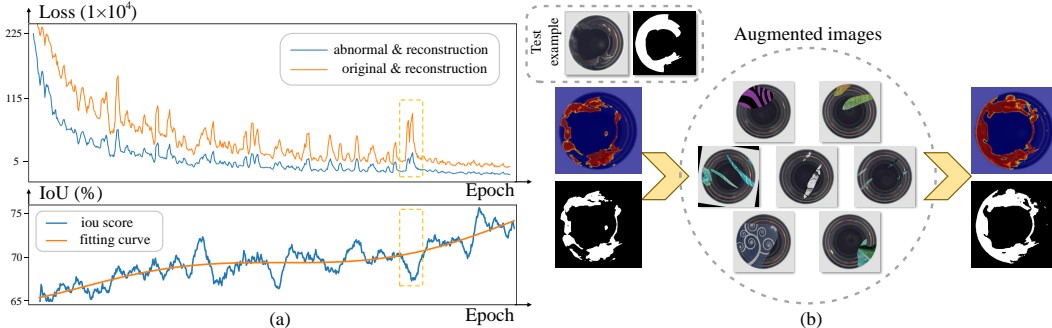

Figure 1: (a) Upper: Segmentation loss curves of the discriminative network. The lower section depicts IoU curves comparing the predicted anomalous masks with synthetic anomalous masks (manually generated ground truth). (b) Upper left: The original test sample and its true anomalous mask (GT), presented as an example to illustrate the motivation behind our method. From left to right: results showing poor model performance, followed by improved performance achieved via our weighted synthesis approach for different regions during anomalous feature learning.

well-reconstructed regions can degrade model performance and even lead to overfitting to synthetic anomalies.

To overcome the limitations, PLSAD introduces a pseudo-label supervised mechanism that generates supervisory signals from synthesized anomalies while maintaining an unsupervised paradigm. This approach enhances the discrimination path between original images and synthetic images, while simultaneously considering the reconstruction quality and the abnormal clues arising from differences with the original images. As indicated by the red curves in the upper half of Figure 1 (a), although the trend resembles that of conventional loss, it reveals greater disparities. By explicitly aligning the reconstruction discrepancies with synthetic anomalies through pseudo-labels, we simulate the pseudo-label guiding mechanism in semi-supervised learning. This helps reduce prediction uncertainty and improve model stability. Moreover, as demonstrated by the yellow dashed box in Figure 1 (a), the segmentation loss and IoU scores exhibit a degree of complementarity. Using this characteristic, we assign higher loss weights to low IoU areas (which indicate poor reconstruction capability), dynamically focusing the model on challenging regions. Figure 1 (b) illustrates the training mechanism of our method, which improves the performance of the model by weighted learning of synthetic anomalies in different regions.

Our main contributions are as follows: 1. We identify and address a critical limitation in reconstruction-based anomaly detection: the lack of direct supervisory signals for discriminative feature learning. Our pseudo-label supervision mechanism provides targeted guidance for distinguishing normal from anomalous regions without requiring expensive manual annotations. 2. We propose PLSAD, a novel framework that integrates pseudo-labels into unsupervised anomaly detection by creating an additional learning path between original and synthetically anomalous images. This dual-path architecture effectively decouples the reconstruction process from discriminative learning, resulting in more robust feature representations, reduced error propagation, and enhanced model stability. 3. We introduce Adaptive IoU Weighting (AIW), a dynamic weighting mechanism that intelligently modulates pseudo-label supervision based on the model's region-specific performance. By assigning higher weights to areas with low IoU scores (indicating poor reconstruction capability), AIW focuses on challenging regions while preventing overfitting to synthetic patterns. 4. Additionally, we provide extensive quantitative and qualitative evidence across multiple industrial datasets (MVTec-AD, MVTec-LOCO, and VisA) showing that our method achieves state-of-the-art performance on challenging tasks, including detection (99.4% image-level AU-ROC on MVTec-AD) and localization (93.5% PRO score).

## 2 RELATED WORK

### 2.1 UNSUPERVISED INDUSTRIAL ANOMALY DETECTION.

Visual detection based on deep learning has achieved significant progress with the aid of supervised learning Kwon et al. (2019); Ruff et al. (2021). However, in real-world industrial scenarios, the

scarcity of defect samples, the high cost of annotation, and the lack of prior knowledge about defects often render supervised methods ineffective. Recently, unsupervised industrial anomaly detection (IAD) algorithms have been increasingly applied to industrial tasks, where the training set contains only normal samples for each category, while the test set comprises both normal and abnormal samples.. Unsupervised IAD methods are primarily categorized into three types: reconstruction-based methods, synthesis-based methods, and embedding-based methods. In this paper, we focus on reconstruction-based methods. Feature embedding-based methods have recently achieved state-of-the-art performance and can be specifically categorized into: teacher-student architecture Bergmann et al. (2020); Deng & Li (2022), normalizing flow Rezende & Mohamed (2015); Rudolph et al. (2021), memory bankRoth et al. (2022); Cohen & Hoshen (2020), and one-class classification Sohn et al. (2020b).

Reconstruction-based methods Haselmann et al. (2018); Ristea et al. (2022); Zavrtanik et al. (2021c) include approaches based on autoencoders Bergmann et al. (2019b); Zavrtanik et al. (2021b); Chen et al. (2023), generative adversarial networks Yan et al. (2022); Duan et al. (2023), Transformers You et al. (2022); Yao et al. (2023), and diffusion models Lu et al. (2023); Zhang et al. (2023). Although these methods have been widely adopted in recent years, they require substantial training time and generally underperform compared to feature embedding-based approaches, thus posing challenges for practical industrial deployment. Many anomaly detection methods rely on image reconstruction and identify anomalies based on the reconstruction error. Autoencoders are commonly employed for this purpose and are often trained using adversarial loss functions.

## 2.2 Pseudo-Label Techniques

Pseudo-labeling, a technique originally rooted in semi-supervised learning, leverages model predictions on unlabeled data as supervisory signals to enhance training robustness Lee et al. (2013). In classical semi-supervised classification, high-confidence predictions are treated as pseudo-labels to expand the labeled dataset Sohn et al. (2020a). However, these methods require partial anomaly annotations or complex iterative refinement, limiting their practicality in fully unsupervised industrial settings.

PLSAD markedly differs from existing methods by incorporating pseudo-label supervision within a purely unsupervised framework. Unlike the implicit error comparison employed in DRAEM, PLSAD explicitly aligns reconstruction discrepancies with synthetic anomalies via pseudo-labels, effectively simulating semi-supervised learning. Moreover, it dynamically optimizes regions exhibiting low reconstruction quality, thereby mitigating the homogenization bias observed in current optimization strategies. Although PLSAD generates artificial labels, it fundamentally operates within a reconstruction-based unsupervised paradigm. While previous pseudo-label methods aimed at increasing noise Im et al. (2025) and improving model distribution generalization Pan et al. (2025), our approach emphasizes targeted optimization within specific latent spaces.

## 3 Method

### 3.1 Model Architecture

Our method is an unsupervised reconstruction-based anomaly detection approach, where the PLSAD architecture, illustrated in Figure 2, comprises a reconstruction network and a discriminative network. The synthesized anomalous image $I_a$ is fed into the reconstruction network, which detects and reconstructs anomalies by generating semantically plausible normal content, while preserving the nonanomalous regions of the input image unchanged. Subsequently, the discriminative network produces an accurate anomaly segmentation map $M_{pre}$ based on the concatenation of the reconstructed and anomalous inputs. Building upon this, we introduce the concatenated normal-anomaly inputs. Since the synthesized anomalies can provide accurate segmentation maps, the unsupervised approach transitions to supervised (though it is a form of pseudo-supervision because the anomalies are artificially generated), resulting in an anomaly segmentation map $M_{ps}$, which is then binarized and compared with the ground truth via IoU.

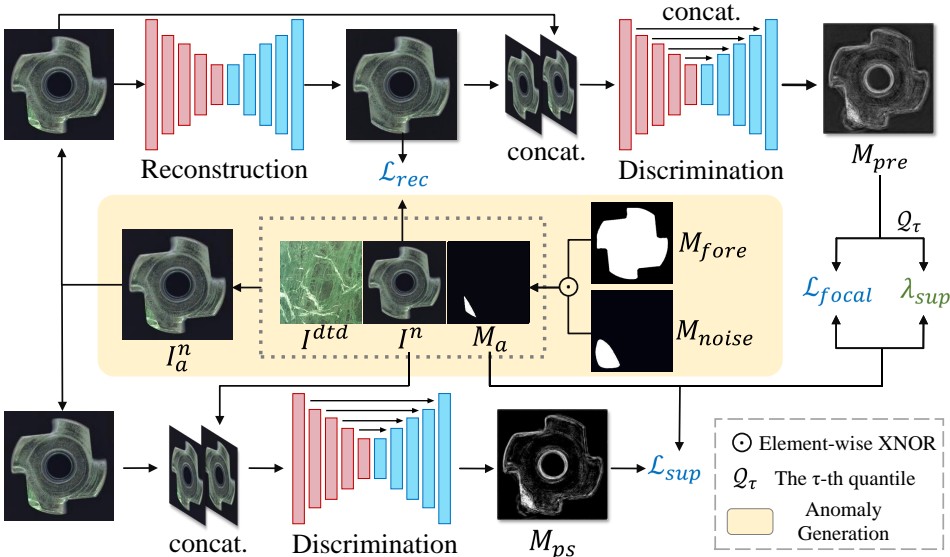

Figure 2: The architecture of PLSAD consists of a reconstruction network and a discriminative network. At the top of the diagram is a generic reconstruction-based process, where the reconstruction network repairs the abnormal synthesized image. Subsequently, the input and output of the reconstruction network are connected and fed into the discriminative network, which outputs the anomalous map $M_{pre}$. Below is the pseudo-label supervised process, where normal images are connected with the abnormal synthesized image, similarly inputting into the discriminative network to produce the anomalous map $M_{ps}$.

### 3.1.1 RECONSTRUCTION AND DISCRIMINATION NETWORKS

**Reconstructive Network**: A U-Net-style encoder-decoder architecture trained to restore artificially corrupted images $I_a$ (generated via synthetic anomaly injection) to their original normal state $I^n$. The reconstruction loss combines pixel-level ($L_2$) distance and patch-based structural similarity (SSIM) Wang et al. (2004) to enforce both global fidelity and local texture consistency:

$$L_{rec}(I^n, I_{rec}) = \lambda L_{SSIM}(I^n, I_r) + (I^n - I_r)^2, \tag{1}$$

where $L_{SSIM}$ is computed as:

$$L_{SSIM}(I^n, I_{rec}) = \frac{1}{N} \sum_{c,w,h} \left(1 - SSIM(I^n, I_{rec})_{(i,j)}\right), \tag{2}$$

with $H \times W$ denoting the image dimensions and $\lambda$ balancing the two terms.

**Discriminative Network**: A U-Net-based module that takes channel-wise concatenated inputs $I_c = [I, I_{rec}]$(original and reconstructed images) and outputs an anomaly score map $M_o$. The sub-network automatically learns a defect-sensitive distance measure through Focal Loss Lin et al. (2017):

$$L_{seg}(M_a, M_{pre}) = -\frac{1}{N} \sum_{c,w,h} \left[(1 - M_{pre})^\gamma M_a \log(M_{pre}) + M_{pre}^\gamma (1 - M_a) \log(1 - M_{pre})\right], \tag{3}$$

where $M_a$ is the synthetic anomaly mask and $M_{pre}$ is the predicted mask.

### 3.1.2 SIMULATED ANOMALY GENERATION

We employ the DTD dataset Cimpoi et al. (2014) as the source of anomalous textures and utilize the Perlin noise generator Perlin (1985) to synthesize noise patterns capturing diverse anomaly shapes. Because noise may randomly spread beyond target regions, we constrain anomalies to plausible areas

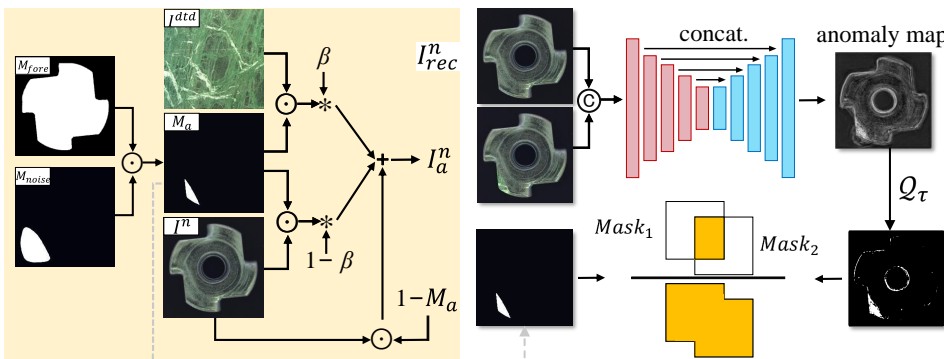

Figure 3: Left: Simulated anomaly generation process. Based on $M_a$, samples are taken from the anomaly texture $I_{dtd}$ and placed on the normal image $I^n$ to generate the anomaly image $I_a^n$, where $\beta$ controls the anomaly region's transparency. Right: Process of $\lambda_{sup}$. The IoU with the abnormal ground truth mask is computed to obtain the reciprocal of $\lambda_{sup}$.

using a foreground-background separation strategy. The anomaly mask $M_a$ is generated by binarizing the synthesized patterns at a uniformly sampled threshold.

Inspired by RandAugment Cubuk et al. (2020), we further enhance textures through stochastic augmentation. The augmented texture image $I^{\text{dtd}}$ and anomaly mask $M_a$ are blended with the original image $I$, as illustrated in Figure 3(a). The augmented training image $I_a$ is formulated as:

$$M_a = M_{\text{fore}} \odot M_{\text{noise}}, \quad I_a = \overline{M_a} \odot I + (1 - \beta)(M_a \odot I) + \beta(M_a \odot I^{\text{dtd}}), \quad (4)$$

where $\overline{M_a}$ denotes the inverse of $M_a$, $\odot$ is element-wise multiplication, and $\beta \in [0.1, 1.0]$ controls the blending opacity.

In our experiments, replacing synthetic textures with the realistic SDAS dataset Zhang et al. (2024) did not improve performance. We attribute this to the advantage of generating near out-of-distribution appearances, which facilitate the model's learning of discriminative distance functions by amplifying the differences between normal and anomalous patterns.

## 3.2 Pseudo-Label Supervised Mechanism

Traditional reconstruction methods only use the concatenation of the reconstructed image and the abnormal synthesized image as the input for the discriminative model. Although this design enables the model to compare repaired and anomalous regions, it inherently ties the discriminative learning process to the performance of the reconstruction sub-network. Any reconstruction errors (e.g., over-smoothing or incomplete inpainting) propagate directly into the discriminative input, thereby introducing noise that destabilizes training.

To address this limitation, we add the concatenation of the original normal image and the abnormal synthesized image $[I^n, I_a^n]$ as input, which also outputs anomaly predictions $M_{ps}$, and we use Focal Loss to compute the discrepancy: $\mathcal{L}_{sup} = L_{seg}(M_a, M_{ps})$, where $M_a$ denotes the synthetic anomaly mask. This approach not only offers a more diverse perspective but also mitigates error propagation by disentangling discriminative learning from reconstruction artifacts, thereby enhancing the stability of PLSAD.

## 3.3 Adaptive IoU Weighting (AIW)

As seen from Figure 1(a), the pseudo-supervisory loss has a significant impact on the overall loss, and how to control this loss is a key factor in achieving better model performance. Therefore, to accommodate the varying regional reconstruction capabilities, we propose dynamically weighting the pseudo-supervisory loss based on the Intersection over Union (IoU). We apply the p-th percentile threshold to binarize the predicted anomaly map $M_{ps}$, yielding the anomaly segmentation prediction $M_{bi}$, and compute the IoU between the two masks as $IOU = \frac{M_{bi} \cap M_a}{M_{bi} \cup M_a}$. IoU effectively reflects the

Table 1: Comparison of PLSAD with different SOTA methods on MVTec-AD datasets. $\cdot/\cdot/\cdot/\cdot$ denotes image-level AUROC%, pixel-level AUROC%, pixel-level AP% and pixel-level PRO%.

| MVTecAD | DRAEM (ICCV 2021) | PatchCore (CVPR 2022) | SimpleNet (CVPR 2023) | EfficientAD (WACV 2024) | RealNet (CVPR 2024) | PLSAD (Ours) |
|---|---|---|---|---|---|---|
| Carpet | 97.0/93.6/54.9/87.7 | 98.7/**99.0**/62.7/94.9 | 99.7/98.2/-/- | 97.4/96.9/**83.0**/91.1 | **99.8**/99.2/-/**97.0** | 99.0/96.5/68.7/93.4 |
| Grid | 99.9/99.3/54.6/**98.0** | 98.2/98.7/32.5/93.9 | 99.7/98.8/-/- | 99.1/97.0/41.3/88.8 | 100.0/99.5/-/97.1 | **100.0**/**99.4**/**66.3**/**98.0** |
| Leather | **100.0**/99.2/**74.7**/**98.1** | 100.0/99.3/45.6/97.4 | **100.0**/99.2/-/- | 86.7/98.3/50.8/97.1 | 99.9/**99.7**/-/96.4 | **100.0**/97.2/63.2/95.3 |
| Tile | 99.6/99.5/96.3/98.0 | 98.7/95.6/54.6/90.6 | 99.8/97.0/-/- | **100.0**/96.5/72.7/88.4 | 99.6/99.3/-/95.3 | **100.0**/99.1/93.1/97.2 |
| Wood | 99.1/96.4/**64.7**/90.7 | 99.2/95.0/47.7/89.4 | 100.0/94.5/-/- | 98.8/93.6/54.3/87.4 | 99.2/**98.2**/-/**91.2** | **100.0**/94.4/64.4/89.6 |
| texture | 99.1/97.6/69.0/94.5 | 99.0/97.5/48.6/93.2 | **99.8**/97.5/-/- | 96.4/96.5/60.4/90.6 | 99.7/**99.2**/-/**95.4** | 99.8/97.3/**71.1**/94.7 |
| Bottle | 99.2/**99.3**/90.9/97.3 | 100.0/98.6/76.8/95.7 | 100.0/98.0/-/- | 100.0/98.6/83.0/94.6 | 100.0/99.3/-/**95.6** | 99.6/98.8/86.8/94.7 |
| Cable | 91.8/94.9/47.5/79.3 | 99.5/**98.4**/65.3/**92.5** | 99.9/97.6/-/- | 95.2/97.3/**65.7**/86.8 | 99.1/98.1/-/90.4 | 96.4/96.1/64.8/86.0 |
| Capsule | 98.5/92.4/39.3/89.8 | 98.1/98.8/44.2/95.8 | 97.7/98.9/-/- | 94.4/98.5/50.4/**96.0** | **99.3**/99.3/-/82.3 | 98.7/96.7/49.2/92.7 |
| Hazelnut | **100.0**/99.4/84.6/**98.5** | 100.0/98.7/53.7/93.8 | **100.0**/97.9/-/- | 99.5/96.1/58.2/91.4 | 100.0/**99.7**/-/93.5 | **100.0**/99.6/**91.4**/**98.5** |
| Metal_nut | 98.7/**99.1**/92.6/96.1 | 100.0/98.4/87.0/91.4 | 100.0/98.8/-/- | 98.4/98.4/90.5/91.9 | 99.7/98.6/-/**96.5** | **100.0**/99.1/**93.7**/96.4 |
| Pill | 98.9/97.2/58.4/88.6 | 96.6/97.4/77.7/94.5 | 99.0/98.6/-/- | 96.8/97.4/79.3/95.9 | 98.3/**99.0**/-/84.4 | 98.4/98.4/78.9/**95.2** |
| Screw | 93.9/96.2/42.0/86.2 | 98.1/99.4/35.4/**96.4** | 98.2/99.3/-/- | 93.7/98.1/38.3/89.9 | 97.7/**99.5**/-/85.2 | **99.6**/98.8/**59.3**/94.0 |
| Toothbrush | **100.0**/97.5/37.6/89.4 | **100.0**/98.7/37.2/91.8 | 99.7/98.5/-/- | **100.0**/98.6/51.1/94.5 | 99.4/98.7/-/90.9 | **100.0**/99.3/68.0/**96.3** |
| Transistor | 93.1/83.4/44.0/73.1 | **100.0**/96.3/61.0/83.7 | **100.0**/97.6/-/- | 99.5/93.6/**71.4**/85.4 | 99.7/98.0/-/86.6 | 99.2/90.4/46.6/82.2 |
| Zipper | **100.0**/98.8/**77.7**/**96.5** | 99.4/98.8/59.5/96.1 | 99.9/98.9/-/- | 95.2/97.5/63.9/91.7 | 99.6/**99.2**/-/88.8 | **100.0**/97.5/68.3/93.1 |
| object | 97.4/95.8/61.5/89.5 | 99.2/98.4/59.8/**93.2** | 99.4/98.4/-/- | 97.3/97.4/65.2/91.8 | **99.3**/98.9/-/89.4 | 99.2/97.5/**70.7**/92.9 |
| average | 98.0/96.4/64.0/91.1 | 99.1/98.1/56.1/93.2 | **99.6**/98.1/-/- | 97.0/97.1/63.6/91.4 | 99.4/**99.0**/-/91.4 | 99.4/97.4/**70.8**/**93.5** |

model's discriminative ability in the anomalous regions; when IoU is large, the model's ability in that region is strong, and vice versa. We leverage this metric to weight the pseudo-supervisory term, implementing it via a simple subtraction: $\lambda_{sup} = 1 - IOU$.

## 3.4 INFERENCE AND LOSS

During inference, the output of the discriminative network is a pixel-level anomaly detection mask $M_o$, which can be directly used for image-level anomaly score estimation, i.e., to determine whether there is an anomaly in the image. Firstly, $M_o$ is smoothed using a mean filtering convolutional layer to aggregate local anomaly response information. The final image-level anomaly score $\eta$ is computed by taking the maximum value of the smoothed anomaly score map:

$$\eta = max(M_o * f_{sf \times sf}), \tag{5}$$

where $f_{sf \times sf}$ is a mean filter of size $sf \times sf$, and $*$ denotes the convolution operation. The training loss consists of three parts: reconstruction loss, segmentation loss, and pseudo-supervision loss, expressed as: The total training objective jointly optimizes both sub-networks:

$$
\begin{aligned}
\mathcal{L} &= \mathcal{L}_{rec} + \mathcal{L}_{seg} + \lambda_{sup}\mathcal{L}_{sup} \\
&= \lambda L_{SSIM}(I^n, I_{rec}) + (I^n - I_{rec})^2 + L_{focal}(M_a, M_{pre}) + \lambda_{sup}L_{focal}(M_a, M_{ps}).
\end{aligned}
\tag{6}
$$

## 4 EXPERIMENT

### 4.1 EXPERIMENTAL SETUP

**Datasets.** MVTec-AD Bergmann et al. (2019a) comprises 5,354 high-res images from various domains, with 3,629 anomaly-free training images and 1,725 test images (both normal and abnormal), along with pixel-level annotations. MVTec-LOCO Bergmann et al. (2022) offers 3,644 images from five industrial categories with structural anomalies (e.g., scratches, dents, contaminations) and logical anomalies (e.g., misplaced or missing objects). VisA Zou et al. (2022) contains 10,821 high-res images (9,621 normal; 1,200 anomalous) spanning 12 object classes, which are grouped into Complex Structures, Multiple Instances, and Single Instances.

**Evaluation Metric.** We evaluated the performance at both the image-level and the pixel-level, using the Area Under the Receiver Operating Characteristic curve (AU-ROC, AUC) as the primary

Table 2: Comparison of PLSAD with different SOTA methods on MVTec-AD datasets. $\cdot/\cdot/\cdot/\cdot$ denotes image-level AUROC%, pixel-level AUROC%, pixel-level AP% and pixel-level PRO%.

| VisA | DRAEM | PatchCore | SimpleNet | RealNet | PLSAD |
|---|---|---|---|---|---|
| candle | 94.4/97.3/93.7 | 98.6/**99.5**/94.0 | **98.7**/98.8/**94.9** | 95.0/99.0/- | 95.4/97.1/93.1 |
| capsules | 76.3/99.1/84.5 | 81.6/**99.5**/85.5 | 91.7/94.9/90.6 | 88.1/97.6/- | **93.0**/98.5/**92.2** |
| cashew | 90.7/88.2/51.8 | 97.3/98.9/94.5 | 97.0/**99.0**/89.2 | 95.9/97.6/- | **97.4**/92.1/**91.5** |
| chewinggum | 94.2/97.1/60.4 | 99.1/99.1/84.6 | 99.8/97.3/83.4 | **100.0/99.8**/- | 97.8/99.2/**88.8** |
| fryum | 97.4/92.7/93.1 | 96.2/93.8/85.3 | 98.4/91.2/86.9 | 95.3/95.2/- | **98.9/96.6/96.1** |
| macaroni1 | 95.0/99.7/96.7 | 97.5/**99.8**/95.4 | **99.4**/98.9/**98.7** | 98.2/99.7/- | 96.8/99.4/97.9 |
| macaroni2 | **96.2/99.9/99.6** | 78.1/99.1/94.4 | 82.9/97.7/93.9 | 91.8/99.3/- | 95.6/99.7/97.0 |
| pcb1 | 54.8/90.5/74.3 | 98.5/**99.9**/94.3 | **99.5**/99.6/92.7 | 97.1/99.4/- | 96.7/99.2/89.9 |
| pcb2 | 77.8/90.5/83.4 | 97.3/**99.0**/89.2 | **99.5**/97.3/**90.9** | 97.5/97.8/- | 98.7/94.6/83.0 |
| pcb3 | 94.5/98.6/89.9 | 97.9/**99.2**/90.9 | **99.0**/99.2/**92.9** | 97.6/98.4/- | 98.0/95.7/89.6 |
| pcb4 | 93.4/88.0/82.1 | **99.6/98.6**/90.1 | **99.6**/96.7/82.7 | 99.2/**98.6**/- | 99.2/97.7/90.9 |
| pipe_fryum | 99.4/90.9/91.7 | 99.8/**99.1**/95.7 | 99.7/99.0/93.6 | **99.9**/98.6/- | 97.3/97.5/95.3 |
| average | 88.7/94.4/83.4 | 95.1/**98.8**/91.2 | 97.1/97.5/90.9 | 96.3/98.4/- | **97.1**/97.3/**92.1** |

Table 3: Comparison of PLSAD with different SOTA methods on MVTec-LOCO datasets. I-AUC denotes average image-level AU-ROC%

| LOCO | I-AUC |
|---|---|
| DRAEM | 83.0 |
| PatchCore | 81.6 |
| SimpleNet | 77.6 |
| AST | 83.7 |
| EfficientAD | 90.7 |
| PLSAD | **92.7** |

Table 4: PLSAD Module Ablation Experiments. The results for image-level AUROC%, pixel-level AUROC%, and PRO% are reported separately. fg/PLS/AIW denotes constraining anomaly generation to the foreground region, pseudo-label supervision, dynamic IoU weight module.

| fg | PLS | AIW | I-AUC | P-AUC | PRO |
|---|---|---|---|---|---|
|  |  |  | 98.0 | 96.4 | 91.1 |
| ✓ |  |  | 98.1 | 97.0 | 92.3 |
|  | ✓ |  | 99.3 | 95.9 | 90.4 |
| ✓ | ✓ |  | 98.8 | 96.6 | 90.8 |
|  | ✓ | ✓ | **99.5** | 96.3 | 90.5 |
| ✓ | ✓ | ✓ | 99.4 | **97.4** | **93.5** |

metric for quantifying image-level (I-AUC) and pixel-level (P-AUC) performance. To ensure a more equitable treatment of anomaly regions of varying sizes, we employed the Per-Region-Overlap (PRO) metric for anomaly segmentation. And report the pixel-wise average precision metric (AP), which is more appropriate for highly imbalanced classes and in particular for surface anomaly detection, where the precision plays an important role.

**Implementation Details.** The reconstructive sub-network is formulated as an encoder-decoder architecture, and the discriminative sub-network uses a U-Net Ronneberger et al. (2015)-like architecture. Refer to Zavrtanik et al. (2021a) for details. The Describable Textures Dataset (DTD) Cimpoi et al. (2014) is used as the anomaly source dataset. In our experiments, the network is trained for 800 epochs, and the Adam optimizer is used with a learning rate of $10^{-4}$ and is multiplied by 0.1 after 400 and 600 epochs.

## 4.2 MAIN RESULTS

We compared our approach with currently representative unsupervised industrial anomaly detection methods Zavrtanik et al. (2021a); Roth et al. (2022); Liu et al. (2023); Batzner et al. (2024); Zhang et al. (2024); Rudolph et al. (2023) in the literature on the MVTec-AD, MVTec-LOCO, and VisA datasets. We evaluated the detection and segmentation capabilities of the model using multiple metrics, with DRAEM serving as the baseline method. The results of other methods reported in the table are derived from a mix of our own tests and data from other published articles. The best results are highlighted in **bold**, and the second-best results are _underline_.

Table 5: Changing the Weighting Method. The left side shows the results of changing the mapping function, while the right side presents the results of applying different multiples for pseudo-label supervised weighting. We report the average image-level AU-ROC and PRO scores on the MVTecAD dataset, with the scores for specific categories provided in the appendix.

| Weighting | w/o $\lambda$ | iou | inc | dec | $e^{-kx}$ | 1.0 | 1.5 | 2.0 | $\lambda_{sup}$ |
|---|---|---|---|---|---|---|---|---|---|
| I-AUC | 98.85 | 97.45 | 99.06 | 98.94 | 98.84 | 98.85 | 98.91 | 98.88 | **99.38** |
| PRO | 90.86 | 88.77 | 90.91 | 92.44 | 92.35 | 90.86 | 91.91 | 92.44 | **93.51** |

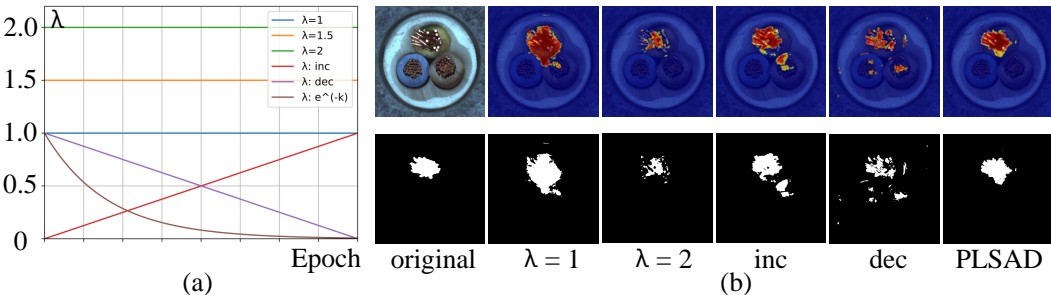

Figure 4: (a) Visualization of different function expressions. (b) Qualitative results of different functions. The first column is the original image and ground truth.

As shown in Table 1, we evaluated PLSAD on the MVTec-AD and presented the image-level AUC, pixel-level AUC, pixel-level AP, and PRO scores for different methods on a per-category basis. PLSAD demonstrated competitive results, improving by 1.4/1.0/6.9/2.4% respectively compared to the baseline, and reducing the error rate by 70/30/20/30%. Particularly in pixel-level evaluations of AP and PRO, it achieved the best scores, indicating that region-based pseudo-label supervision enables the model to enhance its pixel-level capabilities, and more precisely localize and segmentation of anomalous regions.

In Table 2, we evaluated PLSAD on the VisA and presented the image-level AUC, pixel-level AUC, and PRO scores for different methods on a per-category basis. PLSAD exhibits a relatively balanced performance across all categories, with minimal fluctuations in scores between them, indicating that our method is more universally applicable to different types of anomalous objects. Among these, I-AUC and PRO achieved the highest Although PLSAD is not the highest scoring in some datasets and categories, it shows a significant improvement over the baseline scores, reflecting the effectiveness of the pseudo-supervision. In Table 3, we evaluated PLSAD on the MVTec-LOCO and presented the average image-level AUC. Scores for categories will be provided in the appendix. MVTec-LOCO is the most challenging dataset among the three, but it is also the dataset where PLSAD shows the greatest improvement over the baseline, increasing by 9.7%

## 4.3 EMPIRICAL STUDIES

### 4.3.1 EFFECTIVENESS OF DIFFERENT COMPONENTS OF PLSAD

We investigate the effectiveness of each component of PLSAD in Table 4. We presented the image-level AUC, pixel-level AUC, and PRO scores on MVTevc-AD. When the foreground constraint is not used, the image-level score performs best, while the pixel-level score decreases. Both Pseudo Label Supervision (PLS) and Adaptive IoU Weighting (AIW) adjustments can further enhance the results. In summary, the foreground (fg) constraint regulates the anomalous generation areas and reduces background interference. PLS provides explicit supervisory signals, enhancing discriminability, while AIW dynamically optimizes challenging areas, balancing global and local learning. The collaborative effects among these components achieve a performance balance optimization in both image-level anomaly detection and pixel-level localization tasks.

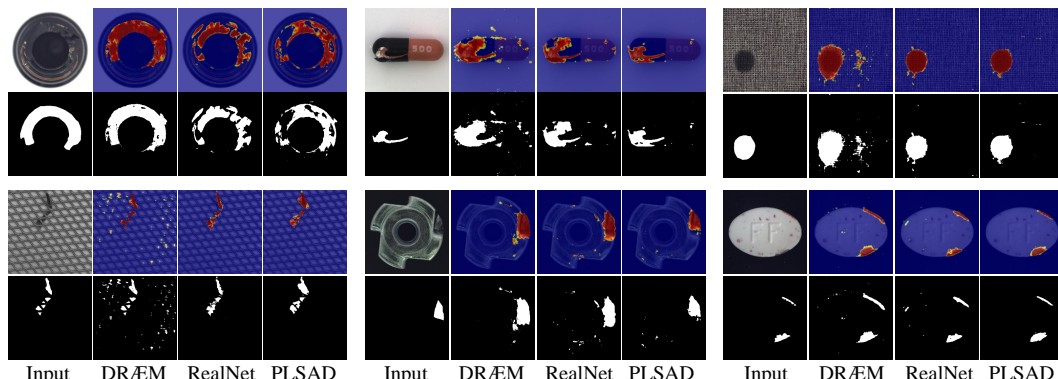

Figure 5: Qualitative results of PLSAD on the MVTec-AD.

### 4.3.2 ADAPTIVE WEIGHTING STRATEGY

As shown in Table 5, we employed different weighting methods to replace AIW, one of which involves changing the dynamic mapping approach, while another uses different multiples of $\lambda_{sup}$.

**Changing Mapping Methods.** We replaced AIW with different function mapping methods, which include: no weighting strategy (w/o $\lambda$), using iou directly (iou), increasing with training iterations (inc), decreasing with training iterations (dec), and inverse Sigmoid function ($e^{-x}$). $\lambda_{sup}$ represents our method. It can be observed that the changes in the image-level metric I-AUC are not significant. However, when using mappings that have a trend significantly different from that of $\lambda_{sup}$, the pixel-level metrics show a noticeable decline. For example, metrics such as IoU and Inc improve as the training epochs increase, but their trends are opposite to that of $\lambda_{sup}$.

**Using Multiple $\lambda_{sup}$.** We replace Adaptive IoU Weighting (AIW) with a static weighting method, setting $\lambda_{sup}$ to a constant value of 1, 1.5, and 2 throughout the training process. As shown in Table 5, we evaluated AU-ROC and PRO on the MVTec-AD dataset and listed the average values (with specific scores for each category provided in the appendix). It can be observed that when the weight of the pseudo-supervisory loss increases, the image-level metric scores slightly increase, but the pixel-level metric scores significantly decrease. We believe that direct comparison of pseudo-supervisory loss provides more global discriminative information, but excessive focus on the original differences can lead to reduced reconstruction performance, which in turn diminishes the model's segmentation capability.

## 4.4 QUANTITATIVE RESULTS

Figure 5 presents the qualitative results of PLSAD on the MVTec-AD dataset. The first column shows the original images and ground truth, the second and third columns display the anomaly map and segmentation mask predictions of DRAEM and RealNet, respectively, and the fourth column shows the results of PLSAD. It can be observed that PLSAD exhibits superior performance.

## 5 CONCLUSION

In this work, we propose PLSAD, which introduces an effective method for unsupervised industrial anomaly detection by integrating pseudo-label supervision into a reconstruction-based anomaly detection framework. The pseudo-supervision mechanism enhances sensitivity to subtle anomalies. And the Adaptive IoU weighting dynamically prioritizes challenging regions, ensuring robust generalization to diverse defect patterns. Experimental results on industrial datasets demonstrate that PLSAD achieves improvements in both detection and localization tasks.

## 6 ETHICS STATEMENT

This work adheres to the ICLR Code of Ethics. In this study, no human subjects or animal experimentation was involved. All datasets used, including MVTec-ADBergmann et al. (2019a),MVTec-LOCO Bergmann et al. (2022) and VisA dataset Zou et al. (2022), were sourced in compliance with relevant usage guidelines, ensuring no violation of privacy. We have taken care to avoid any biases or discriminatory outcomes in our research process. No personally identifiable information was used, and no experiments were conducted that could raise privacy or security concerns. We are committed to maintaining transparency and integrity throughout the research process.

## 7 REPRODUCIBILITY STATEMENT

We have made every effort to ensure that the results presented in this paper are reproducible. All code and datasets have been made publicly available in an anonymous repository to facilitate replication and verification. The experimental setup, including training steps, model configurations, and hardware details, is described in detail in the paper. We have also provided a full description of AIW, to assist others in reproducing our experiments.

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

Table 6: Comparison of PLSAD with different SOTA methods on MVTec-AD datasets. I-AUC, I-AP, P-AUC, and PRO denote image-level AUROC%, image-level AP%, pixel-level AUROC%, and pixel-level PRO%.

| MVTec-LOCO | Baseline | | | | PLSAD | | | |
|---|---|---|---|---|---|---|---|---|
| | I-AUC | I-AP | P-AUC | PRO | I-AUC | I-AP | P-AUC | PRO |
| breakfast_box | 83.65 | 91.35 | 55.89 | 68.87 | 92.30 | 94.29 | 78.55 | 72.68 |
| juice_bottle | 97.02 | 98.95 | 95.21 | 88.37 | 98.70 | 98.83 | 93.65 | 85.32 |
| pushpins | 74.98 | 82.99 | 72.96 | 43.08 | 95.30 | 89.90 | 78.55 | 54.75 |
| screw_bag | 73.90 | 84.02 | 70.79 | 45.29 | 82.34 | 88.94 | 77.57 | 55.68 |
| splicing_connectors | 85.42 | 92.13 | 69.78 | 61.95 | 95.00 | 94.54 | 69.12 | 80.14 |
| average | 82.99 | 89.89 | 72.93 | 61.51 | 92.73 | 93.30 | 79.49 | 69.71 |

Table 7: The full result of changing the mapping function. We report the average image-level AU-ROC and PRO scores on the MVTecAD dataset.

| MVTec-AD | w/o $\lambda$ | iou | inc | dec | $e^{-kx}$ | $\lambda_{sup}$ |
|---|---|---|---|---|---|---|
| Carpet | 98.87/**93.89** | 87.43/79.25 | **99.35**/88.48 | 99.15/93.51 | 97.47/90.48 | 99.03/93.44 |
| Grid | 96.65/95.05 | 99.66/94.49 | **100.0**/96.45 | **100.0**/98.26 | **100.0/98.38** | **100.0**/97.98 |
| Leather | **100.0**/98.42 | **100.0**/96.03 | **100.0**/93.51 | **100.0**/98.52 | **100.0/98.82** | **100.0**/95.65 |
| Tile | **100.0/98.37** | 98.34/88.41 | **100.0**/98.18 | **100.0**/97.86 | 99.92/94.45 | 99.86/97.21 |
| Wood | 99.82/69.99 | **100.0**/81.43 | **100.0**/87.09 | 99.91/**90.43** | **100.0**/81.90 | **100.0**/89.61 |
| Bottle | 99.52/94.31 | 97.06/86.37 | 99.20/83.46 | 97.61/88.92 | 98.65/**95.27** | **99.60**/94.73 |
| Cable | 96.00/82.77 | 90.92/74.95 | 95.53/78.06 | 96.17/79.61 | 94.77/78.15 | **96.38/86.02** |
| Capsule | 98.68/93.68 | 96.41/93.34 | 98.72/93.98 | **98.92/95.03** | 97.44/94.51 | 98.68/92.69 |
| Hazelnut | 99.96/96.94 | **100.0**/96.65 | 99.89/98.39 | **100.0**/98.16 | 99.92/98.08 | **100.0/98.49** |
| Metal_nut | 99.56/95.71 | 99.95/95.84 | **100.0/97.36** | **100.0**/97.15 | 99.90/97.11 | **100.0**/96.42 |
| Pill | 98.39/94.59 | 98.47/94.24 | 97.29/93.48 | 98.69/**95.20** | **99.07**/94.53 | 98.36/95.17 |
| Screw | 98.38/96.30 | 98.85/92.04 | 99.34/95.33 | 95.69/94.20 | 98.77/**97.00** | **99.56**/93.99 |
| Toothbrush | **100.0**/96.80 | 98.88/92.79 | **100.0**/96.40 | **100.0**/94.56 | **100.0**/96.97 | **100.0**/96.28 |
| Transistor | 96.95/65.43 | 96.54/76.57 | 96.54/70.98 | 98.00/71.06 | 96.75/74.88 | **99.16/82.22** |
| Zipper | **100.0**/90.72 | 99.29/89.13 | **100.0**/92.47 | 99.89/94.10 | 99.94/**94.75** | **100.0**/93.10 |
| average | 98.85/90.86 | 97.45/88.77 | 99.05/90.91 | 98.93/92.44 | 98.84/92.35 | **99.37/93.53** |

# A  LIMITATION AND DISCUSSION

**Limitations** Despite its strong performance, PLSAD has several limitations that warrant further investigation. First, the framework's reliance on synthetic anomalies introduces a domain gap between simulated and real-world defects. While the foreground constraint and adaptive weighting mitigate this issue to some extent, subtle or highly context-dependent anomalies (e.g., material fatigue cracks with no visible texture changes) may still evade detection due to insufficient simulation diversity. Second, the computational overhead of the dual-path discriminative sub-network and dynamic IoU weighting could hinder real-time deployment on edge devices, particularly for high-resolution industrial imagery. Third, the foreground constraint assumes accurate object segmentation masks, which may not always be available in practical scenarios with cluttered backgrounds or overlapping objects. Finally, PLSAD's performance on logical anomalies (e.g., missing components in assemblies) remains suboptimal compared to structural defects, as logical anomalies often require higher-level semantic reasoning beyond pixel-level reconstruction.

**Discussion** We illustrate the reason behind the improved model performance of our method through the trend of loss changes throughout the entire training process. As shown in Figure 1(a), the trend of pseudo-supervision is similar to that of reconstruction supervision but more pronounced, which is due to its comparison with the original image, resulting in greater differences. However, overly explicit guidance may cause the model to converge to local optima. Therefore, we require a precise pixel-level adjustment factor to control the strength of pseudo-supervision by predicting the quality of the mask. It is evident that the mask intersection-over-union (IoU) has achieved this goal.

Table 8: The full result of applying different multiples for pseudo-label supervised weighting. We report the average image-level AU-ROC and PRO scores on the MVTecAD dataset.

| MVTec-AD | 1.0 | | 1.5 | | 2.0 | | $\lambda_{sup}$ | |
|---|---|---|---|---|---|---|---|---|
| | I-AUC | PRO | I-AUC | PRO | I-AUC | PRO | I-AUC | PRO |
| Carpet | 98.87 | **93.89** | 97.99 | 91.09 | 95.06 | 88.06 | **99.03** | 93.44 |
| Grid | 96.65 | 95.05 | **100.00** | 98.61 | **100.00** | **98.93** | **100.00** | 97.98 |
| Leather | **100.00** | 98.42 | **100.00** | **98.46** | **100.00** | 97.79 | **100.00** | 95.29 |
| Tile | **100.00** | 98.37 | **100.00** | 96.22 | **100.00** | **98.54** | 99.86 | 97.21 |
| Wood | 99.82 | 69.99 | **100.00** | 91.94 | **100.00** | **93.68** | **100.00** | 89.61 |
| Bottle | 99.52 | 94.31 | 99.04 | 84.16 | 98.41 | 93.49 | **99.60** | **94.73** |
| Cable | 96.00 | 82.77 | 94.60 | 80.67 | **97.60** | 83.27 | 96.38 | **86.02** |
| Capsule | **98.68** | **93.68** | 98.40 | 93.54 | 97.36 | 91.53 | **98.68** | 92.69 |
| Hazelnut | 99.96 | 96.94 | **100.00** | 96.99 | **100.00** | **99.04** | **100.00** | 98.49 |
| Metal_nut | 99.56 | 95.71 | **100.00** | **97.57** | 99.80 | 95.57 | **100.00** | 96.42 |
| Pill | **98.39** | 94.59 | 98.25 | 93.52 | 97.89 | 92.84 | 98.36 | **95.17** |
| Screw | 98.38 | 96.30 | 99.11 | **96.55** | **99.61** | 96.49 | 99.56 | 93.99 |
| Toothbrush | **100.00** | 96.80 | **100.00** | **97.10** | **100.00** | 93.33 | **100.00** | 96.28 |
| Transistor | 96.95 | 65.43 | 96.20 | 67.66 | 97.58 | 76.03 | **99.16** | **82.22** |
| Zipper | **100.00** | 90.72 | 100.00 | **94.58** | 99.94 | 88.06 | **100.00** | 93.10 |
| average | 98.85 | 90.86 | 98.91 | 91.91 | 98.88 | 92.44 | **99.38** | **93.51** |

The trade-offs observed in PLSAD's design highlight inherent challenges in unsupervised anomaly detection. For instance, while foreground constraints improve pixel-level precision, they slightly reduce image-level AUROC by suppressing background false positives—a necessary compromise for industrial applications prioritizing localization accuracy. Similarly, the pseudo-supervision mechanism introduces a delicate balance: overly strict alignment with synthetic anomalies risks overfitting, while insufficient constraints fail to leverage the benefits of explicit guidance. The dynamic weighting strategy, though effective, requires careful calibration of IoU thresholds to avoid over-prioritizing noisy regions. Interestingly, PLSAD's performance on the LOCO dataset (focused on logical anomalies) suggests that reconstruction-based methods inherently struggle with semantic deviations unrelated to texture or structure, where logical anomaly detection demanded hybrid approaches combining reconstruction and symbolic reasoning or introduce auxiliary modules (such as autoencoders). Furthermore, the framework's dependency on synthetic data generation raises questions about its scalability to niche industrial domains with rare or proprietary defect types, where anomaly simulation may lack sufficient prior knowledge.

**Future Directions** To address these limitations, several promising directions emerge: 1. Cross-Domain Synthetic Anomaly Generation: Leveraging diffusion models or physics-based simulators to synthesize anomalies that better mimic real-world defect evolution (e.g., corrosion progression, mechanical wear). 2. Efficient Architecture Design: Exploring lightweight variants of PLSAD, such as replacing the U-Net backbone with vision transformers optimized for edge deployment, or adopting knowledge distillation to compress the dual-path discriminative network.

# B  ADDITIONAL EXPERIMENTS

## B.1  FULL RESULT IN VISA

We provide the complete results on the VisA datasets in Table 6. As shown in the Table, we evaluated the anomaly detection and localization capabilities and presented the image-level AUC, pixel-level AUC, pixel-level AP, and PRO scores for different methods on a per-category basis. PLSAD demonstrated competitive results, improving by 9.73/3.41/6.56/8.20% respectively compared to the baseline, and reducing the error rate by 57/33/24/21%.

## B.2  FULL RESULT OF THE MAPPING FUNCTION

We provide the complete results about Table 5 on the MVTec datasets in Table 7 and Table 8.

Table 9: The results of PLSAD with different reconstructive networks on MVTec-AD, MVTec-LOCO, and VisA dataset. In LOCO and VisA, from left to right, denotes image-level AUROC%, pixel-level AUROC%, and pixel-level PRO%. The average value is marked with a gray background.

| MVTec-AD | I-AUC | I-AP | P-AUC | P-AP | LOCO | 90.9 | 67.6 | 31.7 |
|----------|-------|------|-------|------|------|------|------|------|
| Carpet | 98.4 | 99.5 | 95.4 | 61.8 | breakfast_box | 92.3 | 85.7 | 50.3 |
| Grid | 97.7 | 99.3 | 94.8 | 30.6 | juice_bottle | 98.7 | 83.6 | 42.0 |
| Leather | 100.0 | 100.0 | 99.0 | 68.4 | pushpins | 95.3 | 63.4 | 43.7 |
| Tile | 100.0 | 100.0 | 98.7 | 93.5 | screw_bag | 73.3 | 47.6 | 3.9 |
| Wood | 100.0 | 100.0 | 95.5 | 73.2 | splicing_connectors | 95.0 | 57.8 | 18.7 |
| **textures** | 99.2 | 99.8 | 96.7 | 65.5 | **VisA** | 97.0 | 97.2 | 92.1 |
| Bottle | 99.7 | 99.9 | 98.9 | 89.0 | candle | 95.4 | 97.1 | 93.1 |
| Cable | 95.6 | 97.1 | 94.6 | 54.3 | capsules | 93.0 | 98.5 | 92.2 |
| Capsule | 97.4 | 99.5 | 95.5 | 52.1 | cashew | 97.4 | 92.1 | 91.5 |
| Hazelnut | 100.0 | 100.0 | 99.5 | 86.2 | chewinggum | 97.8 | 99.2 | 88.8 |
| Metal_nut | 100.0 | 100.0 | 99.6 | 97.4 | fryum | 97.8 | 96.6 | 96.1 |
| Pill | 98.4 | 99.7 | 98.0 | 52.3 | macaroni1 | 96.8 | 99.4 | 97.9 |
| Screw | 99.5 | 99.9 | 61.5 | 25.3 | macaroni2 | 95.6 | 99.7 | 97.0 |
| Toothbrush | 99.4 | 99.8 | 98.0 | 50.3 | pcb1 | 96.7 | 99.0 | 89.9 |
| Transistor | 97.7 | 97.6 | 81.0 | 33.4 | pcb2 | 98.7 | 94.3 | 83.0 |
| Zipper | 100.0 | 100.0 | 98.1 | 72.8 | pcb3 | 98.0 | 95.7 | 89.6 |
| **object** | 98.8 | 99.3 | 92.5 | 61.3 | pcb4 | 99.2 | 97.7 | 90.9 |
| **average** | 98.9 | 99.5 | 93.9 | 62.7 | pipe_fryum | 97.3 | 97.5 | 95.3 |

Table 10: Comparison of PLSAD with different SOTA methods on MVTec-AD datasets. I-AUC, I-AP, P-AUC, and PRO denote image-level AUROC%, image-level AP%, pixel-level AUROC%, and pixel-level PRO%.

| MVTec-AD | SDAS+DRAEM | | | | SDAS+PLSAD | | | | PLSAD | | | |
|----------|-------|-------|------|------|-------|-------|------|------|-------|-------|------|------|
| Category | I-AUC | P-AUC | P-AP | PRO | I-AUC | P-AUC | P-AP | PRO | I-AUC | P-AUC | P-AP | PRO |
| Carpet | 98.8 | 95.0 | 66.6 | 90.3 | 97.3 | 92.2 | 50.2 | 85.3 | **99.0** | 96.5 | **68.7** | **93.4** |
| Grid | **100.0** | 99.5 | 68.5 | **98.5** | **100.0** | **99.6** | **70.1** | 98.0 | **100.0** | 99.4 | 66.3 | 98.0 |
| Leather | 99.3 | **98.6** | **70.4** | **97.3** | **100.0** | 98.3 | 68.0 | 96.8 | **100.0** | 97.2 | 63.2 | 95.3 |
| Tile | **100.0** | **99.4** | **95.8** | **98.1** | 96.1 | 96.8 | 86.4 | 89.7 | **100.0** | 99.1 | 93.1 | 97.2 |
| Wood | 98.8 | 93.0 | **68.5** | 87.4 | 98.0 | 90.8 | 59.0 | 80.0 | **100.0** | 94.4 | 64.4 | **89.6** |
| textures | 99.4 | 97.1 | **74.0** | 94.3 | 98.3 | 95.5 | 66.7 | 89.9 | **99.8** | 97.3 | 71.1 | **94.7** |
| Bottle | 98.5 | 95.3 | 84.5 | 92.3 | 97.1 | 95.4 | 73.2 | 86.2 | **99.6** | **98.8** | **86.8** | **94.7** |
| Cable | 94.4 | 92.7 | 62.3 | 82.8 | 92.9 | 93.2 | 61.9 | 72.8 | **96.4** | **96.1** | **64.8** | **86.0** |
| Capsule | 97.3 | **97.3** | **57.1** | **94.5** | 93.3 | 97.1 | 45.4 | 91.1 | **98.7** | 96.7 | 49.2 | 92.7 |
| Hazelnut | **100.0** | 99.1 | 82.6 | 97.1 | 99.9 | 97.8 | 86.2 | 93.5 | **100.0** | **99.6** | **91.4** | **98.5** |
| Metal_nut | 99.8 | 98.0 | 87.7 | **96.2** | 98.4 | 98.5 | 89.8 | 94.2 | **100.0** | **99.1** | **93.7** | 96.4 |
| Pill | 97.2 | **99.2** | **84.4** | **97.1** | 93.1 | 95.0 | 61.6 | 87.6 | **98.4** | 98.4 | 78.9 | 95.2 |
| Screw | 96.3 | **98.8** | **59.7** | **95.3** | 74.6 | 96.1 | 58.1 | 88.1 | **99.6** | 98.8 | 59.3 | 94.0 |
| Toothbrush | **100.0** | 99.2 | 67.6 | 95.4 | 98.9 | 98.5 | 53.9 | 94.3 | **100.0** | **99.3** | **68.0** | **96.3** |
| Transistor | 92.6 | 86.0 | 42.3 | 73.8 | 98.2 | 73.6 | 30.7 | 71.4 | **99.2** | **90.4** | **46.6** | **82.2** |
| Zipper | 99.6 | **98.4** | **75.3** | **94.3** | 98.8 | 95.9 | 53.1 | 86.1 | **100.0** | 97.5 | 68.3 | 93.1 |
| object | 97.6 | 96.4 | 70.3 | 91.9 | 94.5 | 94.1 | 61.4 | 86.5 | **99.2** | **97.5** | **70.7** | **92.9** |
| average | 98.2 | 96.6 | **71.5** | 92.7 | 95.8 | 94.6 | 63.2 | 87.7 | **99.4** | **97.4** | 70.8 | **93.5** |

Table 7 shows the results of changing the mapping function, while Table 8 presents the results of applying different multiples for pseudo-label supervised weighting. We report the average image-level AU-ROC and PRO scores.

### B.3 USE MORE REALISTIC ANOMALIES

In PLSAD, we use the DTD dataset to synthesize anomalies. This approach is more primitive and straightforward than generative models for creating anomalies. To investigate whether the form of anomalies impacts model performance, we replace the DTD dataset with more realistic anomalies from SDAS Zhang et al. (2024). In Table 10, replacing synthetic textures with the realistic SDAS dataset did not improve performance. We attribute this to the benefits of generating near-out-of-

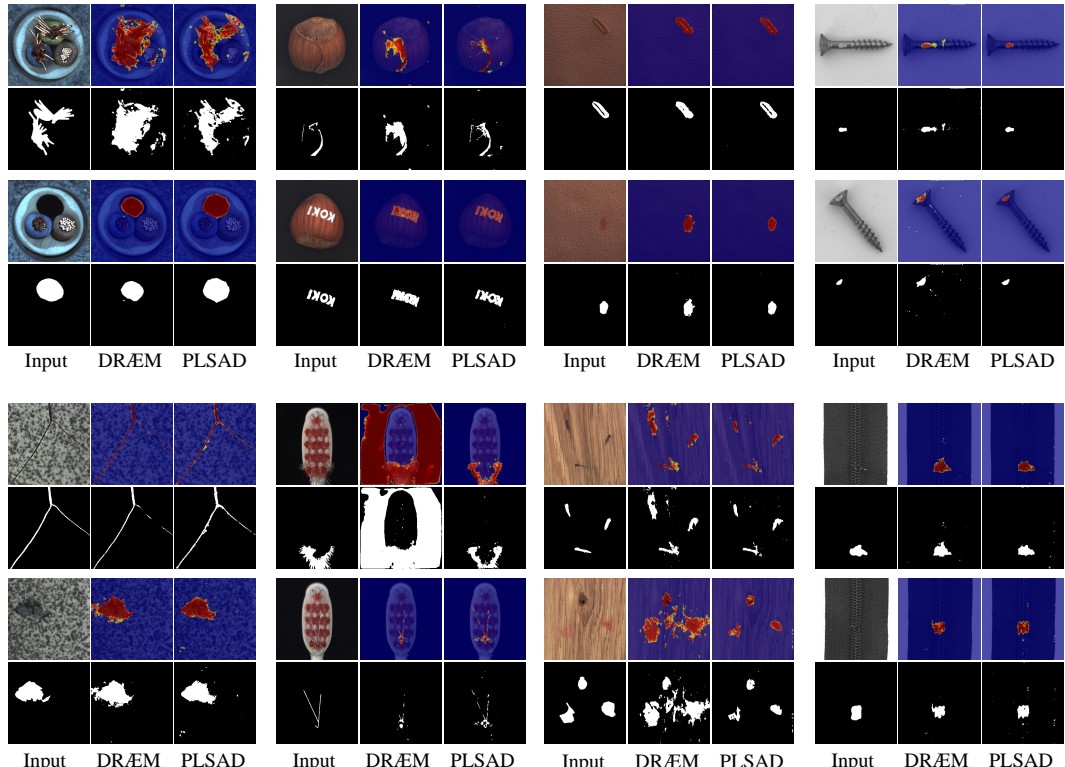

Figure 6: Qualitative results of PLSAD on the MVTec-AD.

distribution appearances, which help models learn discriminative distance functions by amplifying deviations between normal and anomalous patterns.

### B.4 Use Other Reconstruction Models

We utilize the reconstruction model trained on DDAD as a replacement. The choice of DDAD is not only due to its outstanding performance but also because it employs a generative method based on reconstructing random noise, which significantly differs from our approach. As shown in Table 9, the results of I-AUC and PRO on MVTec-AD before and after the replacement are presented.

## C More Qualitative Results

Figure 6 presents the qualitative results of PLSAD on the MVTec-AD dataset. The first column shows the original images and ground truth, the second and third columns display the anomaly map and segmentation mask predictions of DRAEM and PLSAD. It can be observed that PLSAD exhibits superior performance.

---

**Algorithm 1** algorithm of Simulated Anomaly Generation

---

**Input:** training image $I^n$; abnormal texture image $I^{dtd}$; foreground mask $M_{fore}$; poisson noise mask $M_{noise}$; transparency $\beta$;
**Output:** abnormal image $I_a^n$
  1: Intersection of the $M_{fore}$ and $M_{noise}$ , obtain the anomaly mask $M_a$;
  2: Abnormal texture image and abnormal mask dot multiplication: $I^{dtd} \odot M_a$;
  3: Training image and inverse of the anomaly mask dot multiplication: $I^n \odot (1 - M_a)$;
  4: Multiply by transparency and then add: $\beta \left[ I^{dtd} \odot M_a \right] + (1 - \beta) \left[ I^n \odot (1 - M_a) \right]$ **return** $I_a^n$

---

## D  ALGORITHM

The following algorithm is the process is used to obtain abnormal images.

## E  LLM USAGE

Large Language Models (LLMs) were used to aid in the writing and polishing of the manuscript. Specifically, we used an LLM to assist in refining the language, improving readability, and ensuring clarity in various sections of the paper. The model helped with tasks such as sentence rephrasing, grammar checking, and enhancing the overall flow of the text.

It is important to note that the LLM was not involved in the ideation, research methodology, or experimental design. All research concepts, ideas, and analyses were developed and conducted by the authors. The contributions of the LLM were solely focused on improving the linguistic quality of the paper, with no involvement in the scientific content or data analysis.

The authors take full responsibility for the content of the manuscript, including any text generated or polished by the LLM. We have ensured that the LLM-generated text adheres to ethical guidelines and does not contribute to plagiarism or scientific misconduct.

