# OpenReview forum: "Pseudo-Label Supervision in Unsupervised Industrial Anomaly Detection"
_ICLR.cc/2026/Conference — ICLR 2026 Conference Withdrawn Submission_

### Official Review · Reviewer_zyas · 2025-10-22

**Soundness:** 2
**Presentation:** 2
**Contribution:** 3
**Rating:** 4
**Confidence:** 4

**Summary:**

This paper proposes PLSAD, a pseudo-label supervised framework for unsupervised industrial anomaly detection.
By integrating reconstruction and discrimination networks with adaptive IoU weighting, the method introduces explicit supervision from synthetic anomalies, leading to improved detection accuracy and localization.
The study is well-motivated and shows notable improvements across multiple benchmarks, though its overall presentation could be clearer and some figures and tables would benefit from more detailed explanatory context.

**Strengths:**

1.The paper introduces a distinctive idea of reconstructing normal images from anomalous ones and leveraging the reconstruction loss to identify difficult regions. This design provides a novel perspective within industrial anomaly detection, effectively bridging generative reconstruction and discriminative learning.

2.By jointly using both reconstructed normal samples and ground-truth masks for loss computation—combined with the Adaptive IoU Weighting (AIW) mechanism to emphasize hard-to-discriminate regions—the method differs from prior approaches. It makes more efficient use of the reconstructed normal data and achieves measurable performance improvements.

**Weaknesses:**

1.The paper’s presentation and figure/table descriptions could be more polished for clarity.

2.The observed performance improvements are moderate compared to existing baselines.

3.The benefit of using reconstructed pseudo ground truth is not fully articulated.

4.The relationship between the reconstruction process and the AIW mechanism could be discussed in greater depth.

**Questions:**

1.In Figure 1, the caption’s reference to “upper left” or “upper” appears inconsistent or ambiguous. It’s unclear which subfigure these terms refer to, and the visual explanation could be made more precise.


2.In Table 2, the header seems to have been mistakenly copied from Table 1, and the meaning of the three reported metrics in the columns is difficult to interpret. Clarifying what each metric represents would improve readability.

3.The method employs reconstructed normal samples as pseudo ground truth for supervision. If these reconstructed samples are not used for the AIW module, what advantages remain compared to directly using the ground-truth mask alone? Does this design effectively enhance the diversity and coverage of the training samples?

4.Since AIW relies on reconstruction quality to assign adaptive weights, could the reconstruction module be trained separately to provide more stable or generalizable weighting signals? Would such decoupling affect convergence or performance?

---

### Official Review · Reviewer_Nn1c · 2025-10-29

**Soundness:** 2
**Presentation:** 2
**Contribution:** 1
**Rating:** 2
**Confidence:** 4

**Summary:**

In this paper, the author proposed a method that involves the reconstruction residual into the training process with the syntactic pseudo-label (or generated negative sample from the dtd anomaly texture library). It outperforms existing self supervised IAD methods, such as reconstruction based methods to some extent.

**Strengths:**

- The reported results are fairly good

**Weaknesses:**

The overall novelty is low. From my side, it is not qualified for ICLR publication.

- The syntactic pseudo-label is very common in IAD research.  See DREAM and its following papers.

- The authors pointed out the limitation of reconstruction based method. However, the proposed method still utilizes AE.  The fundamental limitations caused by AE still exist. It still uses the reconstruction error to make prediction. I think the authors should explain why their method is more helpful theoratically.

- The authors introduce AIW but they only defined AIW in 3.3. I cannot find which part of the method use AIW

**Questions:**

- The paper title in the system and the submiss are not the same. Which one is correct?

- If some anomaly patterns are not similar to texture in DTD, or some anomaly patterns are similar to the normal part, how the proposed method performs? E.g. in some vision AD task, such as skin, ceramic tile, etc, the anomaly pattern are very small.

- What are the image level F1 and AP results? They are not reported.

---

### Official Review · Reviewer_UBYY · 2025-10-30

**Soundness:** 3
**Presentation:** 3
**Contribution:** 2
**Rating:** 2
**Confidence:** 4

**Summary:**

This paper proposes PLSAD, an unsupervised industrial anomaly detection framework built on reconstruction methods. The main idea is to generate synthetic anomalies and use their masks as pseudo-labels to provide explicit supervision for a discriminative segmentation network. An additional contribution is an Adaptive IoU Weighting (AIW) strategy, which increases loss weights for poorly reconstructed regions to focus training on difficult areas. Experiments on MVTec-AD, MVTec-LOCO, and VisA show improvements over DRAEM and other reconstruction-based baselines.

**Strengths:**

The motivation is clear. Reconstruction-based methods lack explicit discriminative supervision, and pseudo-label training is a reasonable attempt to strengthen abnormal region learning.

The method does not require complex architecture changes and can be plugged into existing reconstruction pipelines.

Ablation studies are fairly thorough, including evaluation of pseudo-labeling, foreground constraints, and different weighting strategies.

Consistent performance gains over DRAEM and similar reconstruction baselines. The improvements on pixel-level metrics (AP/PRO) indicate better localization

**Weaknesses:**

The paper mostly compares against older reconstruction-based techniques (e.g., DRAEM, RealNet). The field has recently shifted toward feature-embedding and foundation-model-based approaches, such as Dinomaly, INP-Former, InCtrl, etc. Without these comparisons, it is difficult to claim that PLSAD achieves competitive state-of-the-art performance.

Pseudo-label supervision and weighted losses have been explored in recent works (e.g., Dinomaly, CDO)The method seems more like a refinement of DRAEM rather than a fundamentally new pipeline.

Image-level AUROC on MVTec-AD is already near 99–100% for many recent methods. Improvements shown here are relatively small, and in some categories PLSAD does not outperform more modern approaches from the literature. Datasets such like Real-IAD, MANTA, M2AD should be involved.

**Questions:**

Could the pseudo-label supervision negatively overfit to simulated patterns, reducing robustness to unseen anomalies?

Why not directly incorporate representation learning from large pretrained vision models like Dino2, which have shown strong zero-shot anomaly localization?

---

### Official Review · Reviewer_SXQm · 2025-10-31

**Soundness:** 2
**Presentation:** 3
**Contribution:** 2
**Rating:** 2
**Confidence:** 3

**Summary:**

This paper proposes PLSAD, a pseudo-labele supervised learning framework for unsupervised industrial anomaly detection. To improve feature robustness, PLSAD introduce a dual-stream architecture that separates reconstruction and discrimination. A novel Adaptive IoU Weighting (AIW) mechanism dynamically highlights regions with poor reconstruction.

**Strengths:**

1. This paper effectively identifies and clearly presents the key limitations of existing reconstruction-based anomaly detection approaches, supporting them with experimental evidence. The motivation for proposing a new framework is sufficiently justified.
2. The authors propose several complementary loss functions that comprehensively improve the generation and reconstruction of anomaly images, contributing to more effective anomaly localization.

**Weaknesses:**

1. This paper does not include an experimental comparison with recent related work in unsupervised anomaly detection.
For example,
1-1. Dinomaly: The Less is More Philosophy in Multi-Class Unsupervised Anomaly Detection (CVPR 2025)
1-2. Towards Real Unsupervised Anomaly Detection via Confident Meta-Learning (ICCV 2025)
Including these baselines would strengthen the empirical validation.

2. Limited novelty and pseudo-labeling analysis
2-1. The pseudo-labeling mechanism lacks clear novelty. For instance, AnomalyGPT: Detecting Industrial Anomalies using Large Vision-Language Models (AAAI 2024) also generates pseudo anomalies using Perlin noise and concatenates them with input images for model training.
2-2. This paper does not include ablation studies evaluating the contribution of the pseudo-labeling module or alternative synthetic anomaly generation strategies beyond the DTD dataset.
2-3. Since pseudo-labeling is presented as a major contribution of this paper, it would be important to apply similar pseudo-labeling techniques to other baseline methods for a fair comparison.

3.This paper lacks any discussion or quantitative analysis of computational efficiency, such as inference time or model complexity, which are essential for practical application in industrial anomaly detection environments.

**Questions:**

1. Is there a valid reason for not including comparisons with the state-of-the-art methods?
2. What are the key differences between the process of synthesizing anomalous images and its application in AnomalyGPT?
3. Can pseudo-labeling techniques be applied to other baseline methods for a fair comparison?

---

### Note · Authors · 2025-11-12

I have read and agree with the venue's withdrawal policy on behalf of myself and my co-authors.